# Bivalve Shellfish Safety in Portugal: Variability of Faecal Levels, Metal Contaminants and Marine Biotoxins during the Last Decade (2011–2020)

**DOI:** 10.3390/toxins15020091

**Published:** 2023-01-18

**Authors:** Ana Catarina Braga, Susana Margarida Rodrigues, Helena Maria Lourenço, Pedro Reis Costa, Sónia Pedro

**Affiliations:** 1S2AQUA—Collaborative Laboratory, Association for a Sustainable and Smart Aquaculture, Av. Parque Natural da Ria Formosa s/n, 8700-194 Olhão, Portugal; 2IPMA, I.P.—Portuguese Institute of the Sea and Atmosphere, Av. Dr. Alfredo Magalhães Ramalho, 6, 1495-165 Lisboa, Portugal; 3CIIMAR—Interdisciplinary Centre of Marine and Environmental Research, Av. General Norton de Matos S/N, 4450-208 Matosinhos, Portugal; 4CCMAR—Centre of Marine Sciences, Campus of Gambelas, University of Algarve, 8005-139 Faro, Portugal

**Keywords:** marine biotoxins, faecal contamination, metal contamination, shellfish, seafood safety, Portuguese coast

## Abstract

Bivalves are a high-value product whose production has markedly increased, reaching 9863 tonnes in Portugal in 2021. Bivalves’ habitats—lagoons, estuaries and coastal waters—are exposed to biological and anthropogenic contaminants, which can bioaccumulate in these organisms and pose a significant public health risk. The need to obtain a safe product for human consumption led to the implementation of standardised hygiene regulations for harvesting and marketing bivalve molluscs, resulting in routine monitoring of bivalve production areas for microbial quality, metal contaminants, and marine biotoxins. While excessive levels of biotoxins and metal contamination lead to temporary harvesting bans, high faecal contamination leads to area reclassification and impose post-harvest treatments. In this study, the seasonal and temporal variability of these parameters were analysed using historical data generated by the monitoring programme during the last decade. Moreover, the impact of the monitoring program on bivalve harvesting from 2011 to 2020 was assessed. This program presented a considerable improvement over time, with an increase in the sampling effort and the overall program representativeness. Finally, contamination risk, revising control measures, and defining recommendations for risk mitigation measures are given in the light of ten years’ monitoring.

## 1. Introduction

Shellfish, particularly bivalve molluscs, have been important in the human diet since ancient times, playing an essential role for human settlements in coastal regions and their prehistoric economies. Previously assumed to be an abundant and self-renewing marine resource [1,2], wild-capture shellfisheries are nowadays limited to the highly exploited natural seed banks without the opportunity to develop or increase catch rates. Due to their considerable nutritional value, and variety of health benefits, there is a growing demand for seafood and seafood products. Aquaculture production, including shellfish farming, has been a growing sector, developing faster than any other animal-food sector. Bivalve farming is seen as a good route to respond to the needs of an increasing human population, estimated to reach 10.9 billion people in 2100 [3].

Farming of mussels, oysters, and clams, among other bivalve shellfish species, has been expanding worldwide in recent decades. In Portugal, bivalve shellfish production is a socioeconomic pillar of many coastal communities. The widespread cultivation of filter-feeding species has been associated with multiple benefits regarding ecological goods and services [4,5]. As with other marine aquaculture productions and in contrast with land animal protein production, bivalve farming does not require any source of freshwater. In the case of filter-feeding species, no added feed is even required, as those organisms take up their diet from the plankton available in the water column. Because of their filter-feeding mechanism, farming of bivalves may strongly contribute to attenuating ocean eutrophication with a positive impact on water transparency [4,5]. Therefore, cultivating mussels and other shellfish species combined with fish farms in an Integrated Multi-Trophic Aquaculture has been a strategy to implement more sustainable systems [6,7].

However, it is precisely their mode of feeding, which is so important and environmentally friendly, that, under certain circumstances, raises concerns regarding bivalve safety for human consumption. Threats to bivalve shellfish salubrity can originate from several sources, from domestic or agro-industrial pollution to natural sources, such as telluric metals or marine biotoxins. Since bivalves have an impressive filtering capacity, they can accumulate to a great extent any contaminant in the water column. Therefore, these invertebrates may pose a significant health risk to whoever consumes them [8,9]. In order to identify and control the potential hazards, the bivalve production areas are periodically monitored to assess whether these invertebrates can be harvested and marketed according to the regulatory health standards applied to the live trade for human consumption [10,11].

The microbiological monitoring of bivalve production areas involves the assessment over a period of time of *Escherichia coli* concentrations, expressed as the most probable number (MPN) per 100 g, as a marker of faecal contamination, and can indicate the public health risk from microbial pathogens. The European Union (EU) Member States have a system for grading bivalve production areas into three classes (A, B or C) based on the increasing faecal (*E. coli)* levels present in the flesh and intravalvular liquid of these animals [11]. This classification grade dictates what level of post-harvesting treatment is needed before live bivalves are placed on the market for human consumption, which can involve purification, relaying or cooking by an approved method [10]. The lack of appropriate monitoring may lead to the consumption of contaminated bivalves, which can cause gastroenteritis and other more severe human illnesses [12].

The surveillance of specific seafood contaminants, such as metal contaminants, including mercury (Hg), cadmium (Cd) and lead (Pb), is performed in order to check their concentrations in relation to regulation EU 2019/627 [11]. For these metallic elements, the EU Member States established a maximum threshold system in which the element has its maximum concentration level defined according to risk assessments and previous contamination events [13]. The ingestion of bivalves containing metal levels above this maximum threshold level can in the long-term cause several chronic diseases in humans and sometimes even lead to death [14,15].

Another regular surveillance is the detection and quantification of marine biotoxin levels in shellfish. This monitoring is often accompanied by the assessment of toxin-producing microalgae presence and density in the water column alerting to the formation of harmful algal blooms (HABs). These surveillances are used to determine the risk associated with biotoxins in each bivalve production area. The maximum threshold system has also been implemented for marine biotoxins [10,16]. Responsible for several food poisoning syndromes, ingestion of marine biotoxins can induce a range of symptoms, from gastrointestinal disorders to neurological problems and even death, depending on the type of biotoxins associated with the acute poisoning in question. The most frequent marine biotoxins on the Portuguese coast are the lipophilic okadaic acid group (okadaic acid—OA, and dinophysistoxins—DTXs) that cause the diarrheic shellfish poisoning (DSP) syndrome; the saxitoxin (STX) and its derivatives associated with the paralytic shellfish poisoning (PSP) syndrome; and domoic acid associated with the amnesic shellfish poisoning (ASP) syndrome. Other lipophilic toxins such as yessotoxins (YTXs) and the azaspiracid (AZA) groups are less frequent in the bivalves from the Portuguese coast [9,10,17,18,19] than elsewhere.

In order to prevent human illnesses, European legislation has laid down regulatory limits (RL), i.e., the maximum legal amounts of microbiological contamination, metal contaminants and marine biotoxins allowed in live bivalves for human consumption, as presented in Table 1. In order to assist in the assessment of compliance by production areas with the microbiological criteria, the RL of class B can be interpreted as the 90th-percentile standard, which is failed if the MPN of *E. coli*/100g is greater than 4600 for more than 10 % of the assessment period and/or if there is a maximum value greater than 46,000. Likewise, the RL of class A can be interpreted as the 80th-percentile standard, which is failed if the MPN of *E. coli*/100g is greater than 230 for more than 20 % of the assessment period and/or if there is a maximum value greater than 700.

Protecting public health and minimising the risk of acute poisonings is the main goal of surveillance of shellfish production areas. However, it may negatively impact producers and other food business operators. In the case of microbial contamination, when classified as B or C, bivalves must be purified/relayed or subjected to industrial processing before being placed in the market [10], representing an additional cost to the producer and the consumer. Microbiological contaminations higher than the limit of class C (*E. col i* > 46,000/100g) can lead to the prohibition of harvesting from the production area for at least three years. Moreover, temporary increased microbiological contamination of shellfish production areas can result in short-term control measures. Besides prohibition of harvesting, such measures may include short-term reclassification and/or increased treatment requirements without reclassification [12].

When monitoring results of the three contaminating metals reveal concentrations below the RL in a species, the harvesting and marketing of this bivalve in that area is permitted and the sampling frequency of two times a year is maintained. In the case where the result of the concentration in at least one of the contaminating metals in a species is equal to or greater than the RL, the harvesting and marketing of this species of bivalve is prohibited, and sampling of this species is intensified to monthly. The ban is lifted for this species after obtaining two consecutive results lower than the RL, which means that harvest is delayed by at least one month. This can also cause additional economic losses to producers.

Regarding biotoxins, when the weekly sampling reveals values above the RL, harvest and commercialisation are prohibited and shellfish in the area is declared toxic. The prohibition extends until toxin levels decrease to below the RL for human consumption for two consecutive results separated by at least 48 h, and only then may shellfish be again exploited. These precautionary measures, which often result in long-term harvest closures, may cause severe economic losses for shellfish producers [23,24], since to-date, no post-harvesting treatment exists to reduce biotoxin contamination.

Several studies have been carried out trying to understand the variability of microbiological contamination [25], the risk of metals [26,27] and the impact of HABs on shellfish toxicity to shellfish consumers [9,28]. Notwithstanding, only a few assessed these hazards’ combined effect on managing bivalve-producing areas [29,30].

In the present study, data from the last decade (2011–2020) on the variability, either temporal or seasonal, of microbiological, metal and biotoxin hazards obtained from the Official Control of the Portuguese bivalve-producing areas (www.IPMA.pt [31]) is reported. The aim of the study is to improve the knowledge on shellfish safety and to provide valuable and comprehensive data/tools to guide and support the activities of stakeholders in the shellfish production chain, such as fisherman/shellfish farmers, other bivalve shellfish business operators, and environmental agencies—as well as consumers themselves. In the context of this paper, shellfish refers to bivalve shellfish.

## 2. Results

### 2.1. The Portuguese Monitoring Programme during the Years 2011 to 2020 

The Portuguese monitoring programme has evolved and changed throughout the last decade to better cope with the demands of shellfish consumers and producers, and to respond to the EU regulation that has been implemented throughout the last decade. This development and upgrade, always focusing on public health safety, included a noticeable intensification of the sampling frequency (Table 2).

The number of samples for microbiological monitoring increased for the majority of the production areas, namely RIAV1, RIAV2, L5, L6 and L8 during the studied period. The highest increase in the number of tested samples was observed in 2014, with rises ranging between 95% and 800%. 

The sampling efforts for metals monitoring in shellfish also improved during the last decade. From 2013 to the present date, at least one species from each production area began to be systematically analysed. 

For marine biotoxins, the sampling effort practically doubled. The exception was L5, where the number of samples received for analysis presented a 24% decrease. The reported reduction resulted from the elimination of one non-strategic sampling point. The shellfish production areas L7 and RIAV1 were the production sites that presented the highest rise in the number of analysed samples, 392% and 503%, respectively. The sampling effort increase in the L7 production area resulted from the two new mussel aquacultures installed in the region in 2014.

### 2.2. Impact on the Shellfish Industry

Microbiological, metal and biotoxin contamination at levels exceeding the EU regulatory safety limits lead to bans and/or restrictions on harvesting and marketing of bivalves from the affected production areas. These measures have an extensive impact on the trade during lengthy harvesting bans. The impact of contamination on the shellfishery industry was assessed by analysing the number of days that harvesting was banned for.

As marine biotoxin monitoring uses a sentinel or indicator species, the presence of biotoxins triggers the harvesting bans for the species existing in the production area. Only after a successful analysis of each individual species is the ban lifted for them [32], thereby potentially increasing the ban impacts. In this study, data from the bans of mussel harvesting was used for the production areas RIAV1, RIAV2, LOB, L5, L6, and L7; and data from donax clam bans were used for production areas L8 and L9. The variability of the number of days closed to harvesting during the decade 2011–2020 is illustrated in Figure 1.

Regarding the estuaries/coastal lagoons production sites, the areas most affected by shellfish closures to harvesting were in the Aveiro lagoon (RIAV1 and RIAV2), leading to a strong history of bans and commonly breaking records of ban periods. This culminated in 2020, with bans reaching 320 and 328 days in RIAV2 and RIAV1, respectively. Óbidos lagoon, on the other hand, is—of all the analysed locations (estuaries/coastal lagoons and littoral areas)—the one with the fewest number of days closed to harvesting, reaching a maximum of 116 closure days in 2020.

As for littoral areas where mussel is the primary indicator/sentinel species for biotoxin contamination, L5 was the area with the highest number of banned days. In this producing area, the number of days closed to harvesting has surpassed 190 days per year since 2015. Littorals L6 and L7 present a lower number of closure days, except for L6 in 2018, where the banned days reached 180 days associated with a rare event of PSP contamination.

In L8 and L9, the number of days closed to harvesting for the indicator species, donax clams, also presented a major variation throughout the decade. In L8, the harvesting bans ranged from 95 days in 2011 to over 200 days per year in 2014, 2015 and 2020. Littoral L9 presents a similar pattern to L8, except for 2017 when the number of banned days decreased in L9, contrary to an increase in L8. The number of banned days per year in L9 is smaller than in L8, ranging from 78 in 2011 to 177 in 2014.

### 2.3. Microbiological Contamination (E. coli)

The eight studied production areas presented a wide range of faecal contamination from 2011 to 2020, without showing a clear temporal trend (Table 3). Production areas RIAV1 and RIAV2 had the highest maximum *E. coli* levels (of 16,000/100g and 36,000/100g) in the years 2019 and 2018, respectively, with maximum faecal concentrations during the decade always well above 700/100g, except in 2013 for RIAV1. Nevertheless, the 90th percentile of RIAV1 and RIAV2 was always below 4600/100g, so these production areas were mainly assessed as class B. At LOB, the highest maximum *E. coli* level of 36,000/100g was measured in 2016, 2018 and 2019, with 90th percentiles of 6540/100g and 5400/100g in 2016 and 2018, respectively. These latter values reflected this production area classification as belonging to class C during the years of 2016 and 2018 (Table 3). 

In relation to coastal areas, the highest maximum faecal contamination was observed in 2015 for L5 (16,000/100g), 2020 for L6 (2400/100g), 2018 for L7 (490/100g), 2014 for L8 (5400/100g) and 2017 for L9 (1400/100g). Meanwhile, L7 presented *E. coli* 80th percentile always below 230/100g, assuring class A levels throughout the decade (Table 3).

Littorals L5 and L8 had maximum *E. coli* levels above 700/100g with 90th percentiles below 4600/100g for most of the years, supporting class B assessments during most of the decade (Table 3). On the other hand, L6 and L9 showed maximum *E. coli* levels below 700/100g with 80th percentiles until 230/100g for most of the years, leading to class A assessments during most of the studied period. 

As for seasonal variation, during the decade, all the estuarine-lagoonar production areas (RIAV1, RIAV2 and LOB) presented the lowest *E. coli* monthly median concentrations during summer months (1.9, 2.2 and 1.6 Log MPN/100g) (Figure 2a–c). Although the highest *E. coli* monthly median levels were registered in winter months (2.6, 2.9 and 2.9 Log MPN/100g, respectively) for these production areas, this season did not present all the maximum values, which were observed in June and April at RIAV1 and RIAV2, respectively, and in February, April, October and November at LOB. With respect to coastal areas, most of them—namely, L5, L8, and L9 (Figure 2d,g–h)—also had the lowest *E. coli* monthly median concentrations during summer months (1.3, <Detection limit (DL) and <DL Log MPN/100g, respectively), whereas L6 and L7 (Figure 2e,f) presented the lowest *E. coli* monthly median levels during other seasons, besides summer. At L5, L6, L8, and L9, the highest *E. coli* monthly median levels (2.4, 1.6, 1.7 and 2.2 Log MPN/100g, respectively) were observed in March, January, October and December, respectively. For L5, the month of March also presented the maximum *E. coli* levels.

### 2.4. Determination of Metals Contaminants (Hg, Cd, Pb) 

Annual results of Pb and Cd for bivalves from the studied production areas are displayed in Figure 3 and Figure 4. The median concentrations obtained indicate low contamination of these two metals in the studied bivalves. These values ranged from DL to 0.55 mg kg^−1^, wet weight, for Pb and from 0.01 to 0.60 mg kg^−1^for Cd. The highest concentrations of Pb were found in the areas of L5 littoral (0.66 mg kg^−1^ in 2017) and RIAV2 estuarine-lagoon (0.61 mg kg^−1^ in 2012); while for Cd, the maximum levels were observed in the two areas estuarine-lagoon RIAV1 and RIAV2 and in the littoral area L7 (0.60 mg kg^−1^ in 2018, 0.40 mg kg^−1^ in 2015, and 0.60 mg kg^−1^ in 2014, respectively). Nevertheless, in general, similar medians were found among studied years for both metals. Only littoral L7 presented a greater variability of Cd concentrations over the years.

Concerning Hg levels, the values were much lower than the RL, and indeed most of the time were lower than the quantification limit (QL)—data shown in the Appendix A. However, the highest concentrations were also registered in estuarine-lagoon areas RIAV1 and RIAV2 and littoral area L5 (0.07 mg kg^−1^ in 2020, 0.04 mg kg^−1^ in 2012 and 0.06 mg kg^−1^ in 2013/2012, respectively). Overall, the median values ranged from QL to 0.03 mg kg^−1^.

### 2.5. Determination of Marine Toxins (Lipophilic, ASP and PSP Toxins)

The harvesting bans illustrated in Figure 1 were almost all due to bivalve contamination with marine biotoxins exceeding the European RL. Considering the groups of marine biotoxins causing interdictions, it is possible to see a marked difference over the decade in the 8 sampling sites (Figure 5). In line with the data from the total number of samples collected (Table 2), L7 is the location with the most samples analysed over time. The exception to this was that the other sites, such as L8 and L9, presented higher sampling incidence in the first 2 years of the study, 2011 and 2012. 

In Figure 5, it is also possible to observe the number of samples exceeding the legal threshold for each group of toxins. DSP toxins are the most frequent on the Portuguese coast, with high toxin concentrations occurring annually throughout the decade (Figure 5a). This phenomenon was more frequent in coastal waters and estuaries than in the Óbidos lagoon, which had a lower incidence of contaminated samples. 

Regarding ASP and PSP toxins, the occurrence seems to be more sporadic, with a reduced incidence compared with DSP toxins. ASP appeared in 2013 and 2015 in the Aveiro lagoon, in 2014 and 2019 in L7, in 2012 in L8, and in 2012, 2013 and 2019 in L9. The events of ASP appeared to be very sparse and short-lived, in that only a very limited number of samples (one or two per year at most) was over the legal limit (Figure 5b).

On the other hand, PSP appeared in the Aveiro lagoon in 2011, 2015, 2016 and 2017, and in 2012 and 2018 in coastal production areas (L6, L7, L8, L9). PSP events seem to be more prolonged over time as the number of samples over the legal limit ranged from 1 to 11 per year (Figure 5c).

The duration of the ban periods depends on the accumulated biotoxin levels and species-specific depuration rates of bivalves. Monthly averages of toxin concentration in the different production areas over the decade are presented in Figure 6 for DSP, which was by far the most prevalent biotoxin on the Portuguese coast. ASP and PSP toxins are also detected in shellfish from the Portuguese coast. ASP occurs every year causing reduced periods of harvesting ban (Appendix A). PSP occurs very irregularly with large periods of absence and then sudden occurrences that can lead to prolonged harvesting bans (Appendix A). In RIAV1 and RIAV 2 (Figure 6a,b), DSP is present throughout the year, with positive events beginning to occur in April and running until November, and DSP monthly averages in mussels being higher than 2 times the RL. DSP concentrations in these two production areas were higher than 10 times the RL in spring months leading to long periods of mussels ban (more than one month).

Despite some DSP bans in the Óbidos Lagoon (LOB) as shown in Figure 5a, the DSP monthly averages observed in mussels are much lower than those observed for RIAV1 and 2 (Figure 6a–c), which led to shorter interdiction episodes (Figure 1). Concerning coastal production areas, all the studied production sites were affected yearly by DSP events between early spring and autumn (Figure 5 and Figure 6d–h). Mussels from L5 presented DSP monthly averages lower than the EU-regulated level, and most of the positive results occurred in the summer months. For L6 and L7, DSP monthly averages were higher than those observed for L5. Mussels from L7 were more affected than those from production area L6 (Figure 6d–f). While most DSP events occurred in the summer for L5 and L6 production areas, for L7, these had a more seasonally widespread occurrence. In some years, the DSP started early in February and were present in all summer months until autumn. Concerning donax clams, data from L8 and L9 presented similar results, with DSP events starting in early spring, and taking place throughout spring and summer months (Figure 6g,h). For these two production areas, monthly averages of DSP in donax clams toxicity were higher than the EU level in all months between April and December, and higher than those observed for mussels from L5, L6 and L7. As a result, the ban periods for donax harvesting were longer than those observed for mussels, especially in the summer months.

Data from monthly averages for ASP toxins are presented in Appendix A. As mentioned above, these events are more sporadic; consequently, monthly ASP averages are lower than the EU RL for all the analysed production areas. The episodes of ASP bans, with ASP monthly averages above the EU RL, occurred only for production areas L7, L8 and L9 and mainly in the spring months.

Regarding PSP toxins, monthly averages are presented in Appendix A. Contamination events relating to PSP toxins on the Portuguese coast were very irregular as well as sporadic in occurrence (Figure 5). Although the presence of PSP toxins seemed to be observed throughout the year, their concentrations were mostly below the RL. The months in which the most intense events occurred seemed to vary along the coast. In the northwest lagoonar production areas, such as RIAV1 and RIAV2, PSP events were more intense in the autumn and winter months, from October to February, while the southern littoral production areas of L7, L8 and L9 presented more intense PSP events during the summer. The PSP event of highest intensity occurred in L5, in October 2018, with PSP levels in mussels over 40 times higher than the safety RL [33].

## 3. Discussion 

Shellfish production is of paramount importance regarding seafood production on the Portuguese coast, representing 58 % of the seafood produced in the country, and amounting to nearly 9863 tonnes in 2021 [34]. Therefore, it is of extreme necessity and interest to assess shellfish contamination events, regardless of their origin, so that they can be better understood, and prevention and minimisation mechanisms may be proposed.

The three major contamination phenomena surveyed and monitored on the Portuguese coast are microbiology, metals and marine biotoxins contamination. 

The metals monitoring program presented a more constant number of samples, with a slight increase in some production areas over the years. Microbiological sampling presented some variability, but with an overall tendency to increase over time The increasing sampling effort for microbiology aimed to comply with the recommendation on basing the maintenance of classification on a dataset of at least 24 results for a three year monitoring period [12].

The biotoxins monitoring program also reflects the progress of the surveillance efforts, for example, on the production area L7, where the development of offshore production sites led, first, to an increase in samplings from 2014 onward; and later, to a restructuring resulting in the subdivision of the production area, with view to improving both offshore and coastal production. Sampling on L5 decreased as a result of the elimination of a non-representative sampling point, and the Minho estuary was also eliminated as a production area since no shellfish harvesting or production occurs in that location [31].

Variability regarding sample frequency is related to the nature of the contaminant in question. While metals can cause a persistent contamination problem, taking a long time to alter their concentrations, microbiological contamination may show a more seasonal variation and be dependent on several factors, from increase in the human population densities and lack of an adequate wastewater treatment, to runoffs from agriculture and livestock industries [35].Marine biotoxins, on the other hand, are dependent on both HABS occurrence and the shellfish species in the area, as toxin accumulation/elimination capacities vary from species to species [36,37,38,39] and some species may retain the toxins for extensive periods [40,41]. 

Notwithstanding the increase in consumer safety, contamination events still caused significant losses to producers over the years. Analysing the number of days that harvests were banned in Portugal, it is possible to see that throughout the decade, all production sites had closures and that, in some cases, these bans covered 90% of the year, 328 days in RIAV1 during 2020 for example. In line with other European regions [29], these values have a significant impact on local bivalve production and commercialisation.

With reference to the microbiological monitoring data from the eight studied production areas, no clear temporal trend was observed in faecal contamination from 2011 to 2020, probably reflecting a lack of improvement in reducing sources of faecal contamination in the vicinity of production areas. Indeed, others have shown that treatment upgrades to continuous discharges and improvements to intermittent discharges are needed for the reduction in *E. coli* levels in shellfish production areas [42]. The estuarine-lagoonar areas registered higher contamination levels than the coastal areas, as expected. Nevertheless, relevant maximum *E. coli* levels were registered in some of the latter areas, such as L5 (16,000/100g), possibly due to punctual malfunction in the treatment of local sewage effluents, showing that coastal areas can be impacted by high levels of faecal contamination. Moreover, for most production areas, summer months showed lower faecal contamination levels, probably associated with lower rainfall and less storm events driven to human sewage discharges [35]. The potential seasonal variations in shellfish faecal contamination should be considered, as this could open up the possibility of a seasonal classification for some production areas [12], defining less contaminated periods—as already established at LOB between 2008–2013 [31]. Such a flexibility in the classification system could lower the economic burden on the shellfish industry.

Regarding the monitoring of metals in the studied areas, there was no ban during this decade. The levels obtained for the three metals in the various species of bivalves were consistently below the limits allowed by the EU. This finding is also observed in other monitoring programs in other countries [43]. However, as could be expected, the three areas presenting the highest values are among those areas with the highest industrial activity studied over the years [44]. This point notwithstanding, remarkable differences were observed among median Cd content in bivalve samples from L7. The high levels of Cd might be associated with human activity (industrial emissions and the application of fertiliser and sewage sludge to farmland) [45]. However, more data would be needed to confirm this finding.

Harmful algal blooms and the marine biotoxins they produce are a major concern to public health safety [9]. The ingestion of these compounds may be so problematic that, in extreme cases, it may lead to the death of the consumer [9]. While shellfish consumption has evolved throughout history and alongside human settlements [1,2], there is still no viable way to induce toxin elimination in shellfish other than to wait for the shellfish to biotransform and eliminate the compounds themselves [46,47]. This process can take from a few days to over a year, depending on the microalgae species, HABs intensity and frequency, shellfish species or other factors that may affect the concentration of biotoxins [48,49]. Therefore, monitoring programs to assess shellfish health and fitness, and prevent consumption of unsafe, contaminated shellfish is vital to ensure public health safety [9,36,46].

The implementation of these safety mechanisms, however, is not simple. Due to the lack of depuration techniques for biotoxin elimination, the only protection method available is the prohibition/ban of shellfish harvesting until the toxin level returns to levels below the legal safety threshold. This leads to severe challenges for the producers as these bans may represent a significant loss in income; moreover, to-date, there is no way to predict the appearance of blooms on the coastline production areas.

During the studied period, DSP toxins—occurring in almost every production area—were shown to be the main cause of bivalve harvesting bans on the Portuguese coast, and lead to long periods of interdictions every year. Considering that DSP episodes begin in the spring and last through the summer until autumn, hence a relatively well established seasonal pattern, forecasting models based on IPMA’s historical data, as presented here, may provide relevant guidance to shellfish producers [50].

With regard to PSP episodes, historical data has shown that despite periods of absence of contamination in bivalves, when PSP toxic events do occur, they are of high intensity, and the highest frequency is in autumn and winter months.

## 4. Conclusions

Overall, during the decade of 2011–2020, the monitoring program of shellfish production areas underwent a considerable improvement, with an increase in the number of tested samples and their representativeness. Concerning classification of shellfish production areas, flexibility in the system implementation, such as reflecting clear and consistent season-dependent variations, could lower the economic burden on the shellfish industry. Additionally, in view of metal levels consistently being below the EU regulatory limits throughout the decade, this contamination issue is unlikely to restrict the shellfish farming/picking activities in the studied areas. On the other hand, the occurrence of biotoxins will certainly persist and an increasing trend is indeed foreseen. Moreover, new and emerging toxins—resulting from changes in climatic conditions, human pressures and nutrient inputs into the coastal areas—are expected to occur on the Portuguese coast, and may represent a threat to seafood safety until they are included in the monitoring program.

Good communication channels and data sharing among institutions are important, as several competent authorities are involved in the management of shellfish waters, production areas and the official control of live bivalves’. 

These competent authorities should implement prevention and remediation measures, including reduction of sources of faecal contamination, pollutant emissions, and environmental restoration when needed, contributing to the socioeconomic development of primary food business operators, high quality and edibility of shellfish products and consumer safety.

The existence of robust and available databases regarding contamination parameters and environmental factors may contribute to a more efficient approach to the problem, promoting the development of predictive models to help forecast and minimise the effects of future contamination events.

The present study can contribute to the development of strategies aiming to minimise the stakeholders’ economic losses and to promote an improvement in the management of shellfish production areas and in shellfish safety on the Portuguese coast. 

## 5. Material and Methods

### 5.1. Classified Shellfish-Producing Areas of the Portuguese Coast

The Portuguese shellfish safety monitoring programme was implemented in 1986 [9,51]. Currently, the Portuguese monitoring programme comprises 40 classified shellfish-producing areas divided into 13 offshore production areas and 27 estuarine and lagoonar areas [31]. Data collected between 2011 and 2020 from Aveiro (RIAV1 and RIAV2) and Óbidos Lagoons (LOB), and the offshore areas L5, L6, L7, L8 and L9 were selected for this study in view of their relevance to shellfish production and impact of HABs (Figure 7).

Several shellfish species are produced in the selected production areas, and different sampling frequencies are used according to the three types of shellfish contamination: microbiological, metals and marine biotoxins, in order to assess the parameters/contaminant levels with regards to regulatory limits (RL) in EU shellfish hygiene regulations, as shown in Table 1. Therefore, different approaches were taken to improve the analysis of the data available regarding each type of contamination. 

Regarding microbiological and metals contamination, the data from the different shellfish species were grouped, by location and year, independently of the species. The implemented sampling periodicity and the species availability decreased the number of samples per species, so a global approach was selected for these parameters. The species used were mainly clams (*Venerupiscorrugata*; *Ruditapesdecussatus*; *Ruditapesphilippinarum*; *Callista chione*; *Spisulasolida*; *Donax* sp.), cockles (*Cerastodermaedule*), oysters (*Crassostrea angulata*; *Crassostrea gigas*; *Ostrea edulis*) and mussels (*Mytilus galloprovincialis*). 

As established in the rules of EU legislation for marine biotoxins monitoring programs, the species with the highest toxin accumulation rate can be used as an indicator for the group of species growing in the same production area [32]. Since 2002, the Portuguese monitoring programme has been using the concept of indicator species for the control of biotoxin contamination in shellfish production areas [9]. In this way, biotoxins data collected for the indicator species for each shellfish production area from 2011 to 2020were used for the present study. For the estuaries/coastal lagoon areas, RIAV1, RIAV2, LOB and littoral areas of the west coast L5, L6, and L7, data from mussels (*Mytilus galloprovincialis*) toxicity were used. For the littoral areas of the south coast L8 and L9, characterised by vast expanses of sandy beaches where natural banks of clams are regularly exploited, the indicator species selected was the donax clam (*Donax*sp.).

### 5.2. Official Control Data

Data from 2014 to 2020 on microbiological contamination, metals, and marine biotoxin contaminants from the classified shellfish-producing areas of the Portuguese coast are available on IPMA—Portuguese Institute for the Sea and Atmosphere—website [31]. 

Data for the period previous to 2014 was obtained from the IPMA’s database archive. All the collected data were obtained through IPMA’s official control programme. The analysis was performed in microbiology, metals and marine biotoxins laboratories—the national official laboratories accredited according to ISO 17025 [32].

The data collected included the microbiological, metals and biotoxins monitoring programme results; the number of samples analysed in the different laboratories per year, independently of species; and, for each production area the number of days that contamination levels exceeded the EU regulatory safety limits, causing harvesting and marketing bans to be officially put in place. The contamination impacts were assessed analysing the mussel harvesting bans in RIAV1, RIAV2, LOB, L5, L6, and L7 production areas, and donax clam harvesting bans in L8 and L9 production areas.

#### 5.2.1. Microbiological Contamination (*E. coli*)

Laboratory representative bivalve samples for *E. coli* testing comprised a minimum of 10 individuals within the normal commercial size range (ISO 6887-3) [52]. Live animals were washed and brushed under potable running water in order to remove material adherent to the shells, drained and dried using absorbent paper. Shells were aseptically opened and the flesh and intervalvar liquid were aseptically extracted and collected into a sterile container, in order to weigh around 100 g. The quantification of beta-glucuronidase-positive *E. coli* was performed using the most probable number (MPN) technique in a five-tube format according to ISO 16649-3 [53], as described in detail by Pedro et al. [54], with a detection limit of 18 MPN/100g. When appropriate, microbiological results were log_10_ transformed.

#### 5.2.2. Determination of Metal Contaminants (Hg, Cd, Pb)

Cadmium (Cd) and lead (Pb) were analysed by graphite furnace atomic absorption spectrometry (Spectr AA-240Z, Agilent) according to NP EN 14084 [55]. Total mercury (THg) was determined by atomic absorption spectrometry (Automatic Hg analyser, AMA 254, LECO) using the methodology described by EPA [56]. Both methodologies are described in detail in [57]. An external calibration method was applied for metals quantification, using commercial standard solutions (1 g L^−1^) for Hg, Cd and Pb (Merck, Darmstadt, Germany). Detection limits for each element were 0.004 (Hg), 0.002 (Cd) and 0.02 (Pb) mg kg^−1^ wet weight; quantification limit (QL) for Hg was 0.011 mg kg^−1^, ww. All analyses were executed in duplicate, and the analytical data for the three metals are reported in mg kg^−1^ of bivalve based on wet weight. To assess the accuracy of the methods, several certified reference materials over the decade were analysed under the same conditions as the samples (DORM-2—dogfish muscle, LUTS-1—Non-defatted lobster hepatopancreas, TORT-2—Lobster hepatopancreas and DORM-4—Fish protein, National Research Council of Canada). The results obtained in this study were always in agreement with the reference values.

#### 5.2.3. Marine Toxins Determination (Lipophilic, ASP and PSP Toxins)

For marine biotoxins testing, a representative sample consisted of circa 1 kg of bivalves, which yielded at least 100 g of soft tissues. These soft tissues were washed to remove detritus and homogenised to give an adequate representation of the toxin concentration of the production area. Testing of EU regulated marine toxins was carried out following the European official reference methods: (1) OA-group toxins, AZAs and YTXs were determined according to the EU-Harmonised Standard Operating Procedure for determination of lipophilic marine biotoxins in molluscs by LC-MS/MS [58,59], and on some occasions, due to equipment failure, OA-group toxin levels were tested by the phosphatase inhibition assay OkatestZeu, which is an alternative EU-recognised detection method for this group of toxins [60]; (2) determination of ASP toxins was carried out following the EU-Harmonised Standard Operating Procedure for determination of domoic acid in shellfish and finfish by RP-HPLC using UV detection [61], with modifications as described in [62]; (3) for determination of PSP toxins, the AOAC Official Method 2005.06 (the so-called Lawrence method)—a liquid chromatographic method with fluorescence detection and pre-column oxidation of toxins—was used as described in [63,64].

All these methods were validated using the certified reference materials, commercially available from the National Research Council Canada for the lipophilic, amnesic and paralytic shellfish groups of toxins. For the identification/quantification of toxins, NRC-certified reference materials were also used on routine analysis and quality control assessment.

For the present study, ASP, PSP and DSP toxins were selected. Not only are they more frequent on the Portuguese coast, but they are also the only data reported for the entire decade by IPMA. The other lipophilic toxins, PTXs, YTXs and AZAs, were excluded from the present study. In the IPMA reports, data were expressed in µg STX eq. kg^−1^ for the PSP, mg DA kg^−1^ for the ASP, and µg OA eq. kg^−1^ for the DSP.

## Figures and Tables

**Figure 1 toxins-15-00091-f001:**
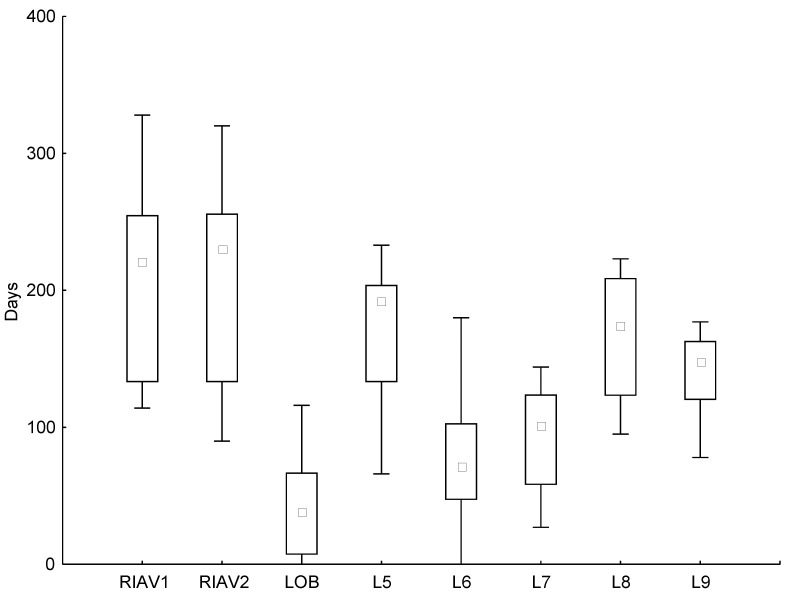
Boxplot of the number of days where shellfish harvesting was banned in each production area between 2011 and 2020. The data presented correspond to the indicator species of each location—*M. galloprovincialis* for RIAV1, RIAV2, LOB, L5, L6, L7, and Donax sp. for L8 and L9. (◻ Median; ⌶ Non-Outlier Range; ○ Outliers; * Extremes).

**Figure 2 toxins-15-00091-f002:**
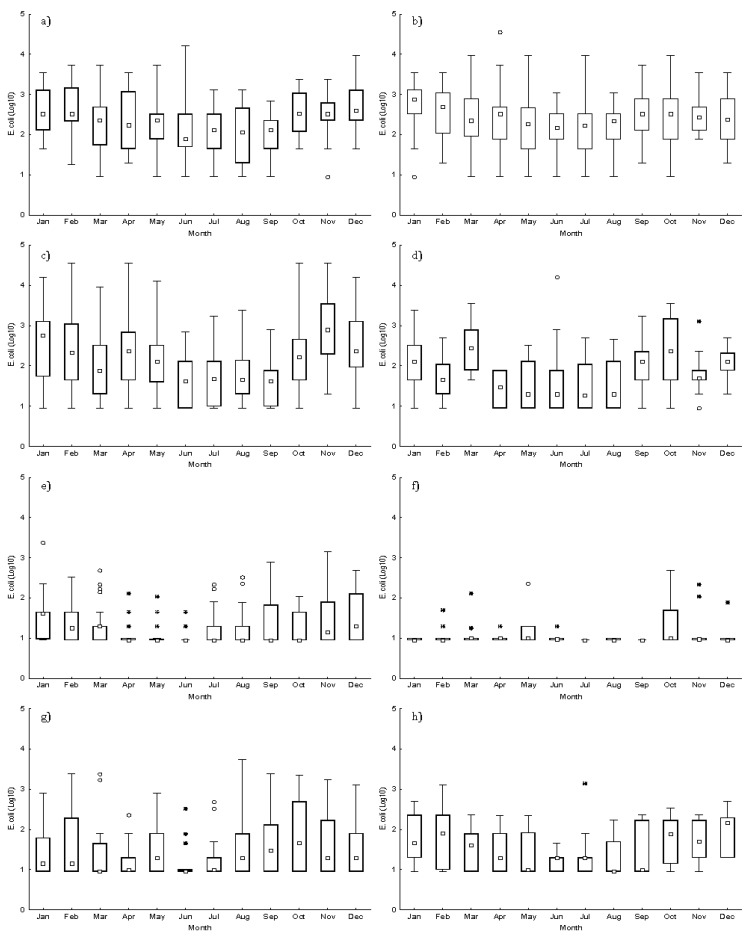
Boxplot of monthly *E. coli* concentrations (Log MPN/100g), between 2011 and 2020, from (**a**) RIAV1; (**b**) RIAV2; (**c**) LOB; (**d**) L5; (**e**) L6; (**f**) L7; (**g**) L8; and (**h**) L9. (◻ Median; ⌶ Non-Outlier Range; ○ Outliers; * Extremes).

**Figure 3 toxins-15-00091-f003:**
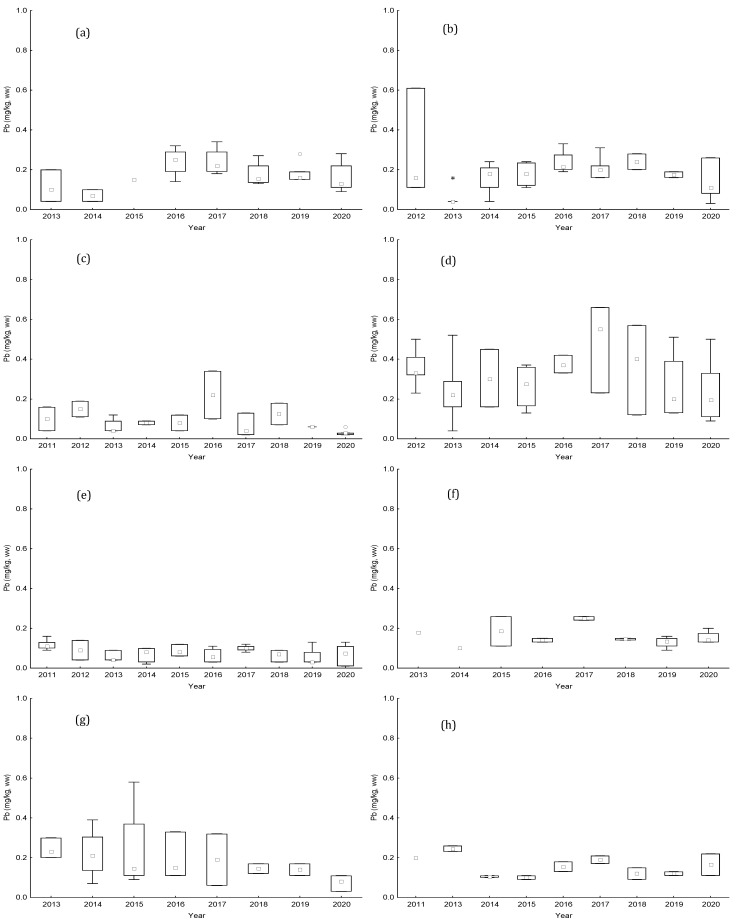
Boxplot of annual lead (Pb) levels (mg kg−1), between 2011 and 2020, from (**a**) RIAV1; (**b**) RIAV2; (**c**) LOB; (**d**) L5; (**e**) L6; (**f**) L7; (**g**) L8; and (**h**) L9. (◻ Median; ⌶ Non-Outlier Range; ○ Outliers; * Extremes).

**Figure 4 toxins-15-00091-f004:**
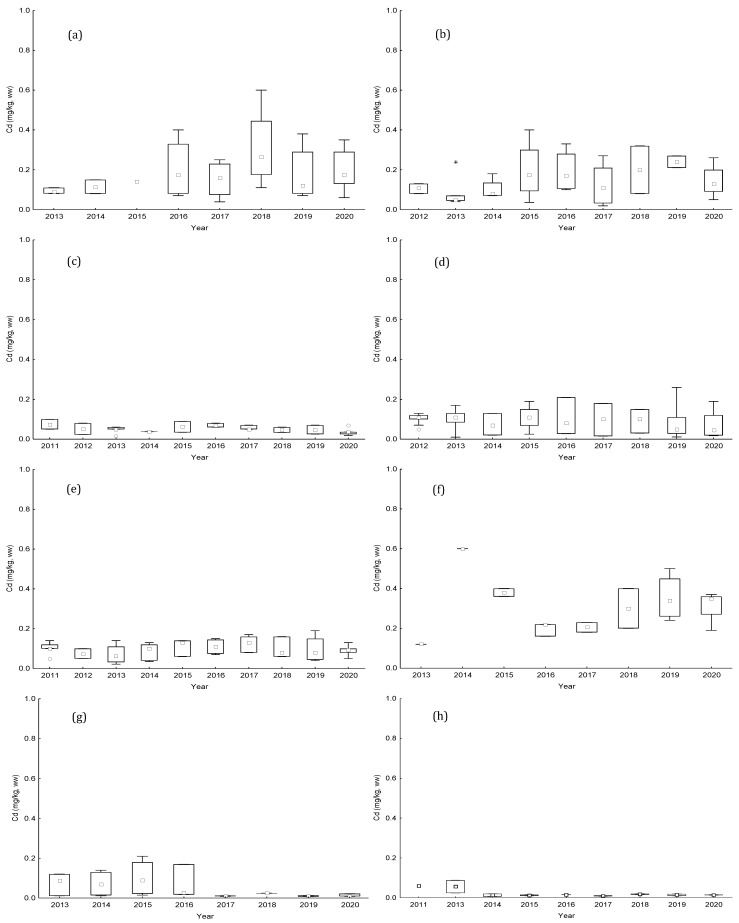
Boxplot of annual lead (Cd) levels (mg kg−1), between 2011 and 2020, from (**a**) RIAV1; (**b**) RIAV2; (**c**) LOB; (**d**) L5; (**e**) L6; (**f**) L7; (**g**) L8; and (**h**) L9. (◻ Median; ⌶ Non-Outlier Range; ○ Outliers; * Extremes).

**Figure 5 toxins-15-00091-f005:**
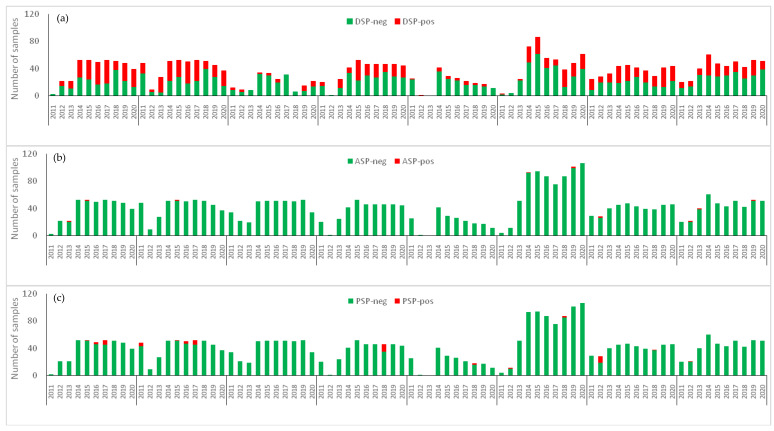
Occurrence of marine biotoxins in shellfish production areas between 2011 and 2020: (**a**) DSP, (**b**) APS, (**c**) PSP. Green indicates the number of samples below the legal threshold while red indicates the number of samples above the legal threshold.

**Figure 6 toxins-15-00091-f006:**
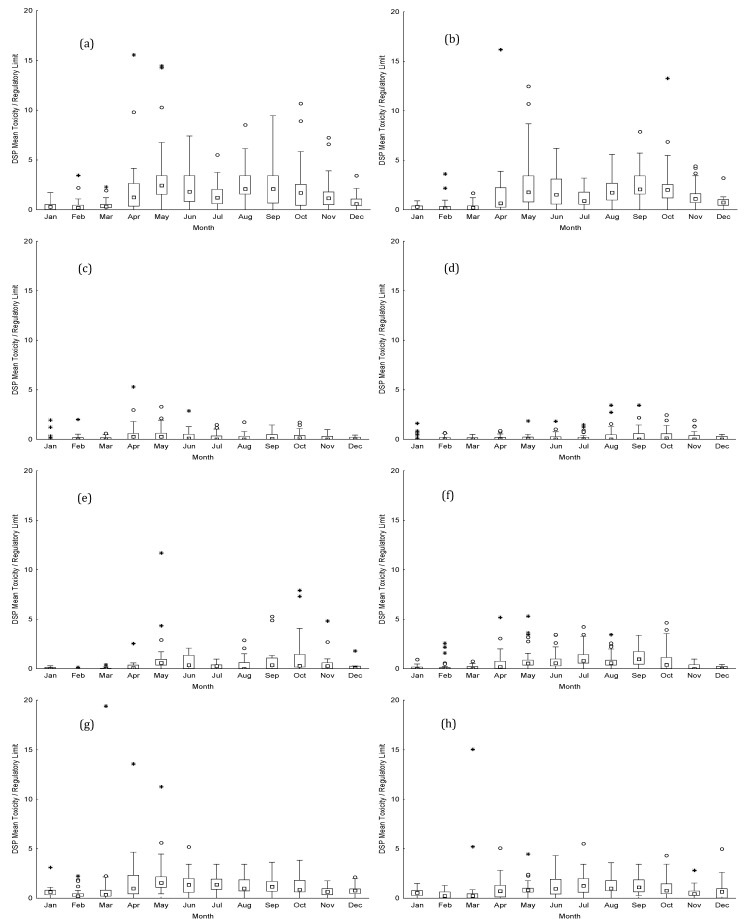
DSP concentration in relation to the EU regulatory limit, between 2011 and 2020, for mussels from: (a) RIAV1; (**b**) RIAV2; (**c**) LOB; (**d**) L5; (**e**) L6; and (**f**) L7—and for donax clams from: (**g**) L8 and (**h**) L9. (◻ Median; ⌶ Non-Outlier Range; ○ Outliers; * Extremes).

**Figure 7 toxins-15-00091-f007:**
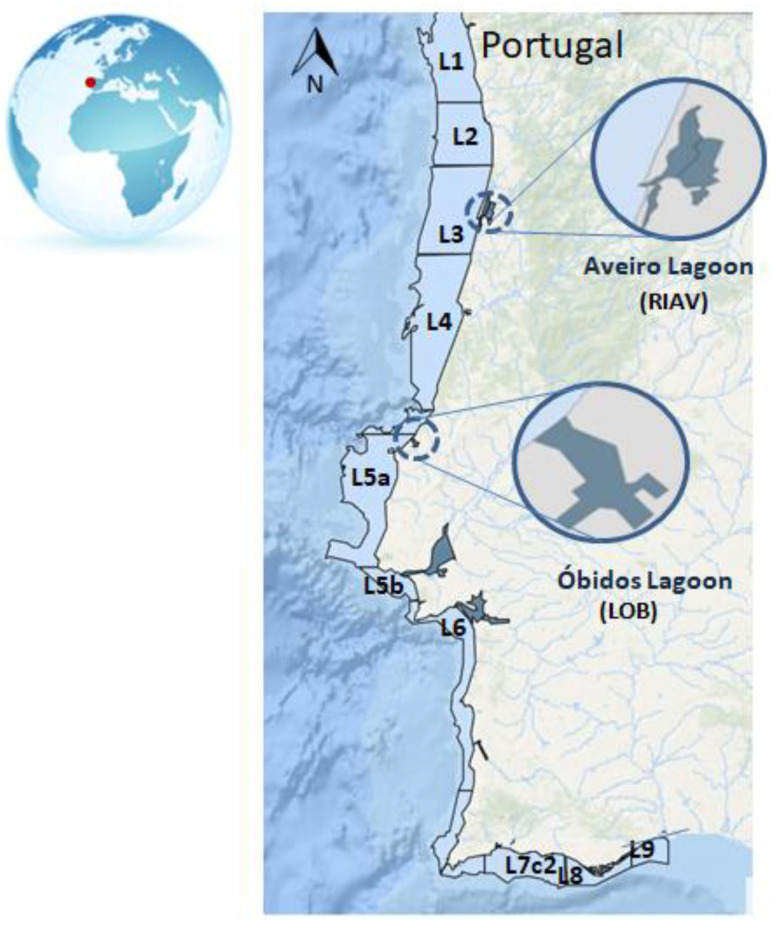
Shellfish production area distribution along the Portuguese coast.

**Table 1 toxins-15-00091-t001:** Legal thresholds of microbiological, metal and marine biotoxin contaminants for bivalve human consumption.

Contaminants	Regulatory Limit (RL)	Reference
**Microbiological ^1^**		
*Escherichia coli*	Class A: 80% of the results ≤ 230 or 2.36 Log (MPN/100g) and 100% of the results ≤ 700 or 2.85 Log (MPN/100g)	[11]
	Class B: 90% of the results ≤ 4600 or 3.66 Log (MPN/100g) and 100% of the results ≤ 46,000 or 4.66 Log (MPN/100g)	
	Class C: 100% results ≤ 46,000 or 4.66 Log (MPN/100g)	
**Metals**		
Mercury (Hg)	0.50 mg kg^−1^	[13,20,21,22]
Cadmium (Cd)	1.0 mg kg^−1^	
Lead (Pb)	1.50 mg kg^−1^	
**Marine Biotoxins**		
PSP toxins	800 µg STX eq. kg^−1^	[10,16]
ASP toxins	20 mg DA kg^−1^	
Okadaic acid toxins group	160 µg OA eq. kg^−1^	
Azaspiracids (AZAs)	160 µg AZA eq. kg^−1^	
Yessotoxins (YTXs)	3.75 mg YTX eq. kg^−1^	

^1^ Live bivalves destined for direct human consumption must also comply with further legal requirements, such as the absence of *Salmonella* in 25 g [10].

**Table 2 toxins-15-00091-t002:** Total number of samples analysed by production area per year for microbiological (“micro”), metal and marine biotoxin (“biotox”) contaminants.

			Year									
			2011	2012	2013	2014	2015	2016	2017	2018	2019	2020
**Shellfish producing areas**	RIAV1	Micro	0	0	3	27	16	28	44	57	38	44
		Metals	0	0	3	2	1	4	4	4	4	6
		Biotox	29	67	108	162	145	135	170	135	164	175
	RIAV2	Micro	0	0	5	30	23	26	47	71	46	44
		Metals	0	3	5	4	4	4	5	2	2	7
		Biotox	108	67	124	161	149	157	169	140	165	149
	LOB	Micro	12	16	15	28	22	28	67	72	51	55
		Metals	2	2	5	2	2	2	3	2	2	6
		Biotox	43	31	39	100	112	118	119	105	110	101
	L5	Micro	0	0	2	14	36	29	24	21	30	17
		Metals	0	10	13	3	4	3	3	3	5	6
		Biotox	102	65	95	88	105	115	24	76	77	78
	L6	Micro	5	2	5	27	28	52	49	37	36	38
		Metals	5	2	6	5	3	4	5	3	4	6
		Biotox	29	17	48	91	111	124	92	108	99	82
	L7	Micro	5	6	6	6	7	8	12	19	22	19
		Metals	0	0	2	2	3	5	2	2	4	4
		Biotox	26	25	74	125	117	94	82	92	109	128
	L8	Micro	13	19	20	39	27	22	27	15	19	24
		Metals	0	0	3	4	4	3	2	2	2	3
		Biotox	59	83	93	124	109	71	53	55	72	63
	L9	Micro	9	7	13	15	10	13	29	16	15	17
		Metals	0	0	2	2	2	2	2	2	2	2
		Biotox	21	22	61	75	55	54	63	55	60	60

**Table 3 toxins-15-00091-t003:** Maximum, 80th and 90th percentiles of *E. coli* levels (MPN/100g) in each shellfish production area per year.

		*E. coli* (MPN/100g)	Year									
			2011	2012	2013	2014	2015	2016	2017	2018	2019	2020
**Shellfish producing areas**	**Riav1**	Max	NA	NA	80	5400	3500	3500	5400	5400	16,000	9200
		80th P.	NA	NA	68	490	330	290	790	1038	856	908
		90th P.	NA	NA	74	1100	410	853	1300	1740	1330	1300
	**Riav2**	Max	NA	NA	2305	9200	1100	3500	3500	36,000	9200	2400
		80th P.	NA	NA	230	790	490	790	490	1100	1300	490
		90th P.	NA	NA	230	956	790	1500	1180	1700	3150	1147
	**LOB**	Max	1300	2400	3500	16,000	9200	36,000	5400	36,000	36,000	14,000
		80th P.	130	790	1520	1036	772	3060	230	1998	490	250
		90th P.	643	1200	3060	1420	2330	6540	330	5400	2400	750
	**L5**	Max	NA	NA	50	330	16,000	790	2400	1700	3500	1300
		80th P.	NA	NA	50	212	560	182	286	490	1340	1100
		90th P.	NA	NA	50	212	560	182	286	490	1340	1100
	**L6**	Max	130	80	80	230	490	790	330	1300	490	2400
		80th P.	42	66	24	71	20	78	20	45	45	45
		90th P.	86	73	52	152	45	302	45	91	78	146
	**L7**	Max	20	50	130	220	170	<DL	230	490	<DL	20
		80th P.	<DL	20	20	220	<DL	<DL	40	<DL	<DL	<DL
		90th P.	16	35	75	220	74	<DL	75	30	<DL	<DL
	**L8**	Max	1700	790	490	5400	170	2200	490	2400	2400	2400
		80th P.	110	80	<DL	<DL	20	45	71	422	790	394
		90th P.	1386	132	49	80	45	77	146	1096	892	790
	**L9**	Max	340	210	230	130	170	170	1400	490	230	1300
		80th P.	170	146	184	80	45	32	58	230	170	230
		90th P.	252	186	228	98	58	112	150	230	200	394

Max—maximum, 80th P.—80th percentile, 90th P.—90th percentile, NA—not applicable, DL—detection limit. Green data—assessed class A, yellow data—assessed class B, orange data—assessed class C.

## Data Availability

Data is available at www.IPMA.pt. (accessed on 26 December 2022).

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
