# Peer review of "Bivalve Shellfish Safety in Portugal: Variability of Faecal Levels, Metal Contaminants and Marine Biotoxins during the Last Decade (2011–2020)"

_toxins, 2023, doi:10.3390/toxins15020091_

Round 1

Reviewer 1 Report

The manuscript provides an interesting study. I think it can be accepted provided some clarifications are carried out.

Abstract

This section could be performed. Too much general information is provided, but few information is given concerning the work carried out and its conclusions. Some more description of the concrete work developed ought to be provided.

Keywords

Portugal coast could be included.

Material and methods

Concerning metal contaminants, some comment on the choice of the three ones selected for the study ought to be given in the text.

Concerning microbial contamination, only E. coli was considered. Some justification on this could be provided in the text.

Conclusions

The need for this kind of studies is especially mentioned in this section. Some conclusions related to the concrete results obtained in the study could be included.

Author Response

The manuscript provides an interesting study. I think it can be accepted provided some clarifications are carried out.

Abstract

This section could be performed. Too much general information is provided, but few information is given concerning the work carried out and its conclusions. Some more description of the concrete work developed ought to be provided.

Author´s response: The abstract was revised

Reviewer #1

Keywords

Portugal coast could be included.

Author´s response: “Portuguese coast” was added as keyword

Reviewer #1

Material and methods

  • Concerning metal contaminants, some comment on the choice of the three ones selected for the study ought to be given in the text.
  • Concerning microbial contamination, only E. coli was considered. Some justification on this could be provided in the text.

Author´s response:

Material and methods section was accordingly revised as follows:

In “5.1. Classified shellfish-producing areas of the Portuguese coast” subsection of Material and methods” it was included the following text: “in order to assess the parameters/contaminants levels with regulatory limits (RL) in EU shellfish hygiene regulations, as shown in Table 1”

So, Lines 186-190 were replaced by: “Several shellfish species are produced in the selected production areas, and different sampling frequencies are used according to the three types of shellfish contamination: microbiology, metals and marine biotoxins, in order to assess the parameters/contaminants levels with regulatory limits (RL) in EU shellfish hygiene regulations, as shown in Table 1. Therefore, different approaches were taken to improve the analysis of the data available regarding each type of contamination.”

Conclusions

The need for this kind of studies is especially mentioned in this section. Some conclusions related to the concrete results obtained in the study could be included

Author´s response:

Conclusions section was revised and improved. The following text was added:

 “Overall, during the decade of 2011-2020, the monitoring program of shellfish production areas presented a considerable improvement, with an increase in the number of tested samples and their representativeness. Concerning classification of shellfish production areas, flexibility in the system implementation, such as reflecting clear and consistent season-dependent variations, can lower the economic burden on shellfish industry. Additionally, obtaining metals levels consistently below the EU regulatory limits, during the decade, suggests that this contamination issue will not restrict the shellfish farming/picking activities in the studied areas. On the other hand, the occurrence of biotoxins will certainly persist and a trend on its increase is foreseen. Moreover, new and emerging toxins, resulting from changes of the climate conditions, human pressures and nutrients inputs into the coastal areas, are expected to occur in the Portuguese coast and may represent a threat to seafood safety until the new biotoxins are included in the monitoring program.”

Reviewer 2 Report

Thank you for sharing your findings on the shellfish quality assurance monitoring in Portugal. The report is of interest to those working in the field. 

Given that summer is the highest risk period for DSP closures, what is the expected impact of climate change on future industry viability? Is this something that should be considered for future industry planning. 

Note: on page 18, line 128 - 'weight' should be change to 'wait'.

Author Response

Reviewer #2

Thank you for sharing your findings on the shellfish quality assurance monitoring in Portugal. The report is of interest to those working in the field.

Given that summer is the highest risk period for DSP closures, what is the expected impact of climate change on future industry viability? Is this something that should be considered for future industry planning.

Author´s response: This is an interesting topic and is the focus of several studies that are currently being perform by this team and will be presented elsewhere. Some clues were already found. DSP toxins as the remaining marine toxins are secondary metabolites, this means they are produced when cells energetic demands are not needed any more for growth. With seawater warming we and several other authors have found that cells take advantage of the conditions to growth, so algal blooms may be more intense with higher cell densities, but it does not means that toxin cell content may increase. We foresee a new balance between HABs occurrence and intensity and shellfish toxicity. This new balance must be characterized and the impact on shellfish closures needs to be assessed.

Note: on page 18, line 128 - 'weight' should be change to 'wait

Author´s response: It was corrected, thank you. Other typos were also revised such as on:

  • page 5 lines 179-180 “potentially” instead of “potencialy”
  • page 15 line 10 “runs” instead of “runns”

Reviewer 3 Report

As a whole i consider the work properly presented and even relevant for quality and management aspects. I suggest its publication and the spreading even in other outreach contexts since the topic is in line with the ongoing SDGs of the 2030 Agenda. This kind of work in my opinion must to be shared, beside to the scientific community itself, even with a wider audience of different stakeholders. Human health and wellbeing are a public matter so these kind of work are very welcome. Congratulations to the authors. 

1. What is the main question addressed by the research?

This contribution is an insight on ecotoxicological aspects related to shellfish farming in Portugal. It analyzes the relevance of pollutants and biotoxins in the target organisms investigated.

2. Do you consider the topic original or relevant in the field? Does it
address a specific gap in the field?

The topic in general is not completely new, since there is a wide extant literature on this field of research, anyway this article can be considered as an overview with a long-term perspective summarizing past and recent data obtained over time on the same issue and relevant for future management and risk mitigation perspectives. It is a valuable contribution due to the final purpose of the study: related to the human health-related needs and to assess the food safety before the distribution of products into the market. It also highlights recent adaptations to the monitoring-related needs

3. What does it add to the subject area compared with other published
material?
An interesting aspect is that beside the classical microbiological analyses an insight on the EU recognized toxicological regulation in force with a specific reference to the grading system of bivalve production areas. Therefore, on this perspective the study, beside the aforementioned research questions, acquires a higher potential for a wider range of implications even at higher level of investigation and for more effective environmental management and long-term monitoring perspectives.

4. *What specific improvements should the authors consider regarding the
methodology? What further controls should be considered? *

In my opinion due to the relevance of the study and according to the FAIR data principles, beside the academic publishing field (which often is not aimed at a general public and risks to remain confined into a too confined world of specialists), the authors should consider a distribution of their insights and data on extant digital platforms such as the EU-supported infrastructures aimed also at a general public and based on more easily consultation rules. The outreach mission is part of the research duties so I would warmly encourage them to consider this perspective in the near future.

5. *Are the conclusions consistent with the evidence and arguments presented
and do they address the main question posed?*

In general, the conclusions are consistent with the main question posed. Maybe a deeper insight on the climate-driven toxic events can be provided to the reader and in order to explain the occurrences of this rising level of toxic compounds some data on climatic aspects referred to the study area can be useful also on a mitigation perspective. Considering that the management aspect is addressed in the aims of this work as currently conceived. In addition, also other speculations on the microbiological aspects investigated could be addressed in order to better explain the real cause behind the data observed during the lab analysis.

6. Are the references appropriate?
As a whole the mentioned references are appropriate, but just before the final submission have a look if more recent works have been published. If so, please integrate the references of your work for a more exhaustive result.

7. Please include any additional comments on the tables and figures.
Figure 7 please provide a georeferenced map instead of a simple figure, showing the coordinated along the figure margins and giving the detail of Portugal position in the Mediterranean context through a smaller square on the right top angle.
The graphs and other tables are properly showed, no additional suggestions.

Author Response

Reviewer #3

As a whole i consider the work properly presented and even relevant for quality and management aspects. I suggest its publication and the spreading even in other outreach contexts since the topic is in line with the ongoing SDGs of the 2030 Agenda. This kind of work in my opinion must to be shared, beside to the scientific community itself, even with a wider audience of different stakeholders. Human health and wellbeing are a public matter so these kind of work are very welcome. Congratulations to the authors.

  1. What is the main question addressed by the research?

This contribution is an insight on ecotoxicological aspects related to shellfish farming in Portugal. It analyzes the relevance of pollutants and biotoxins in the target organisms investigated.

  1. Do you consider the topic original or relevant in the field? Does it address a specific gap in the field?

The topic in general is not completely new, since there is a wide extant literature on this field of research, anyway this article can be considered as an overview with a long-term perspective summarizing past and recent data obtained over time on the same issue and relevant for future management and risk mitigation perspectives. It is a valuable contribution due to the final purpose of the study: related to the human health-related needs and to assess the food safety before the distribution of products into the market. It also highlights recent adaptations to the monitoring-related needs

  1. What does it add to the subject area compared with other published material?

An interesting aspect is that beside the classical microbiological analyses an insight on the EU recognized toxicological regulation in force with a specific reference to the grading system of bivalve production areas. Therefore, on this perspective the study, beside the aforementioned research questions, acquires a higher potential for a wider range of implications even at higher level of investigation and for more effective environmental management and long-term monitoring perspectives.

  1. *What specific improvements should the authors consider regarding the methodology? What further controls should be considered? *

In my opinion due to the relevance of the study and according to the FAIR data principles, beside the academic publishing field (which often is not aimed at a general public and risks to remain confined into a too confined world of specialists), the authors should consider a distribution of their insights and data on extant digital platforms such as the EU-supported infrastructures aimed also at a general public and based on more easily consultation rules. The outreach mission is part of the research duties so I would warmly encourage them to consider this perspective in the near future.

  1. *Are the conclusions consistent with the evidence and arguments presented and do they address the main question posed?*

In general, the conclusions are consistent with the main question posed. Maybe a deeper insight on the climate-driven toxic events can be provided to the reader and in order to explain the occurrences of this rising level of toxic compounds some data on climatic aspects referred to the study area can be useful also on a mitigation perspective. Considering that the management aspect is addressed in the aims of this work as currently conceived. In addition, also other speculations on the microbiological aspects investigated could be addressed in order to better explain the real cause behind the data observed during the lab analysis.

Author´s response: We highly appreciated the kind comments of this reviewer. Suggestions to improve speculations on the microbiological aspects were addressed as follows:

On Introduction section, page 3, lines 114-116 were completed with:

  • “Moreover, temporary increased microbiological contamination of shellfish production areas can result in short-term control measures. These measures can include, besides prohibition of harvesting, short-term reclassification and/or increased treatment requirements without reclassification [12].”

On Discussion section,

  • page 17, lines 108-109 were completed with: as L5 (16000/100g), possibly due to punctual malfunction in the treatment of local sewage effluents, showing that coastal areas can be impacted by high levels of faecal contamination.
  • page 18, lines 109-111 were changed by: “Moreover, for most production areas, summer months showed lower faecal contamination levels, probably associated with lower rainfall and less storm events driven to human sewage discharges [35]. The potential seasonal variations in shellfish faecal contamination should be considered, as this can open up the possibility for seasonal classification of some production areas [12], defining less contaminated periods, already established at LOB between 2008-2013 (IPMA.pt [31]). This flexibility in the classification system can lower the economic burden on shellfish industry.”

So, Conclusions section was revised and improved as follows:

  • On page 18, lines 153-154: “Overall, during the decade of 2011-2020, the monitoring program of shellfish production areas presented a considerable improvement, with an increase in the number of tested samples and their representativeness. Concerning classification of shellfish production areas, flexibility in the system implementation, such as reflecting clear and consistent season-dependent variations, can lower the economic burden on shellfish industry. Additionally, obtaining metals levels consistently below the EU regulatory limits, during the decade, suggests that this contamination issue will not restrict the shellfish farming/picking activities in the studied areas. On the other hand, the occurrence of biotoxins will certainly persist and a trend on its increase is foreseen. Moreover, new and emerging toxins, resulting from changes of the climate conditions, human pressures and nutrients inputs into the coastal areas, are expected to occur in the Portuguese coast and may represent a threat to seafood safety until the new biotoxins are included in the monitoring program.”

  1. Are the references appropriate?

As a whole the mentioned references are appropriate, but just before the final submission have a look if more recent works have been published. If so, please integrate the references of your work for a more exhaustive result.

  1. Please include any additional comments on the tables and figures.

Figure 7 please provide a georeferenced map instead of a simple figure, showing the coordinated along the figure margins and giving the detail of Portugal position in the Mediterranean context through a smaller square on the right top angle.

The graphs and other tables are properly showed, no additional suggestions.

Author´s response: We highly appreciated the kind comments of this reviewer. Figure 7 was improved as suggested.